# Mitigation of Heat Stress in *Solanum lycopersicum* L. by ACC-deaminase and Exopolysaccharide Producing *Bacillus cereus*: Effects on Biochemical Profiling

**Tehmeena Mukhtar [1,2], Shafiq ur Rehman [3], Donald Smith [2], Tariq Sultan [4], Mahmoud F. Seleiman [5,6], Abdullah A. Alsadon [5], Amna [1], Shafaqat Ali [7,8,*], Hassan Javed Chaudhary [1,*], Talaat H. I. Solieman [5,9], Abdullah A. Ibrahim [5] and Montasir A. O. Saad [5]**

[1]  Department of Plant Sciences, Quaid-i-Azam University, Islamabad 45320, Pakistan; tehmeena.mukhtar14@gmail.com (T.M.); amna_qau@yahoo.com (A.)
[2]  Plant Science Department, McGill University, 21,111 Lakeshore Road, Ste. Anne de Bellevue, QC H9X 3V9, Canada; donald.smith@mcgill.ca
[3]  Department of Botany, University of Okara, Okara 53900, Pakistan; evergreenpk@gmail.com
[4]  Land Resource Research Institute, NARC, Islamabad 44000, Pakistan; Tariqsultannarc@gmail.com
[5]  Plant Production Department, College of Food and Agriculture Sciences, King Saud University, P.O. Box 2460, Riyadh 11451, Saudi Arabia; mseleiman@ksu.edu.sa (M.F.S.); alsadon@ksu.edu.sa (A.A.A.); talaat.solieman@yahoo.com (T.H.I.S.); adrahim@ksu.edu.sa (A.A.I.); montysaad26@gmail.com (M.A.O.S.)
[6]  Department of Crop Sciences, Faculty of Agriculture, Menoufia University, Shibin El-kom 32514, Egypt
[7]  Department of Environmental Science and Engineering, Government College University, Faisalabad 38000, Pakistan
[8]  Department of Biological Sciences and Technology, China Medical University, Taichung 40402, Taiwan
[9]  Vegetable Crops Department, Faculty of Agriculture, Alexandria University, P.O. Box 21527 Alexandria, Egypt
*   Correspondence: shafaqataligill@yahoo.com (S.A.); hassaan@qau.edu.pk (H.J.C.)

**Abstract:** Soil microorganisms might be assessed for their capabilities of plant growth promotion in order to identify heat tolerant strategies for crop production. The planned study was conducted to determine the potential of heat tolerant plant growth promoting rhizobacteria (PGPR) in mitigating heat stress effects in tomato. *Bacillus cereus* was evaluated for plant growth promoting activities and assessed for 1-aminocyclopropane-1-carboxylate (ACC-deaminase) (0.76–C0.9 µM/mg protein/h), and exopolysaccharide (0.66–C0.91 mg/mL) under normal and heat stressed conditions. Plant growth regulators were evaluated through High Performance Liquid Chromatography. Bacterial inoculation effects on important physiological and biochemical parameters were evaluated under normal and heat stressed conditions in growth chamber. The morphological-physiological traits significantly revealed drastic effects on both of un-inoculated tomato varieties under heat stress conditions. Bacterial augmentation significantly promoted shoot, root length, leaf surface area, fresh and dry weight. Heat stress enhanced extracellular polymeric substances (EPS) production and cleavage of ACC into a-ketobutyrate and ammonia due to ACC-deaminase producing bacteria that significantly reduced the adverse effects of heat on tomato growth. In conclusion, the applied plant growth promoting rhizobacteria (PGPR) bacterial strain proved as potential candidate for improving tomato crop growing under heat stressed conditions. However, it is highly suggested to validate the current results by conducting field trials.

**Keywords:** ACC-deaminase; heat tolerance; rhizobacteria; PGPR; plant growth regulators; tomato

## 1. Introduction

Global warming is a major threat to living organisms, and has become a critical issue worldwide. Abiotic stresses, such as high temperature, droughts, flash floods, cold waves, elevated carbon dioxide ($CO_2$) and cyclones are natural disasters which can cause economic losses and provide the proof of global warming [1–3]. Global circulation models gave the prediction that greenhouse gases will become the major reason for steadily increasing the average ambient temperatures around the world, and mean temperature per decade of the world will rise by 0.3 °C resulting in temperature increases of approximately 1 and 3 °C in 2025 and 2100, respectively. Heat stress is a problem to the agriculture field and there is an imperative need to tackle this problem for sustaining high productivity of crop plants under high temperatures [4].

High temperature is a major environmental concern that constrains vital plant functions such as seed germination, seedling growth, plant metabolism, and reduces its yield in various agro-ecological zones throughout the world [5,6]. However, elevated temperature has a strong impact on crop yield that varies with different severity levels and duration of heat stress [7,8]. Seed germination may be delayed or inhibited due to high temperatures at 30 to 38 °C [9]. Particularly, reproductive stage of plants has been found to be more sensitive for heat stress, as reported in many crops, such as chickpea [10], lentil [11], mung bean [12], wheat [13] and sorghum [14]. Proline is an amino acid that accumulates in plants under different abiotic stresses such as heat, drought, cold, heavy metals, nutrients, and salt stress; and it can play a beneficial role in growth and flowering of plants [15,16]. Disruption of proline transport and sugar metabolism occurs during the narrow window of male reproductive processes under elevated temperature that cause the failure of fruit setting in tomato plants [17].

In agricultural practices, the application of beneficial microbes is an integral component which should be validated to enhance crop productivity in a defensible way under different abiotic stresses [18]. Plant growth promoting rhizobacteria (PGPR) assist the plant growth either by direct mechanisms which include the production of plant growth regulators, enhanced nutrient availability or by indirect mechanisms which encompasses the suppression of pathogens by antibiosis, induced systemic resistance (ISR) and synthesis of lytic enzymes [19]. During abiotic stress, plant growth promotion activities have been reported in cucumber [20], maize [21], tomato [22], mung bean [23], white clover [24] and wheat [25]. PGPR improved growth of plants by increasing the uptake of nutrients, particularly mineral phosphorus [19,26]. Phytohormones production like gibberellic acid, indole-3-acetic acid, cytokinins, abscisic acid, and antibiotics and siderophore play vital roles in this regard [27,28]. PGPR produce antioxidants that enhance the abscisic acid (ABA) accumulation and degradation of reactive oxygen species [29]. Bacteria such as *Pseudomonas* survive under stress conditions due to exopolysaccharides production [30]. This mechanism provides defense to microorganisms under abiotic stress conditions [31]. ACC producing bacteria have the ability to supply the nitrogen and energy to plants [32]. Inoculation with ACC-deaminase producing bacteria induced longer roots and provided help in the taking up of more amounts of water under stress conditions that, in turn, increased the efficacy of the plants under abiotic stress conditions [33].

Tomato is one of the most economically significant and widespread horticultural crops ranked 7th position in the world [34,35] and its production was 34 million tons in 2018 [36]. Tomato plants displayed a diversity of microbial habitats in the microbial hotspot of the rhizosphere. The discrimination between rhizosphere and root communities of mature plants from unplanted soil profiles is on the basis of selective enrichment of individual bacterial members of microbiota. These level of enrichments displayed a bias for members of phyla *Bacteroidetes*, *Actinobacteria*, *Proteobacteria* (including classes Alpha-, Beta-, Delta-, and *Gammaproteobacteria*) and *Verrucomicrobia*. Members of these taxa have been reported in studies that focused on plant-competent bacteria under both control and field conditions [37].

Several bacterial strains isolated from the tomato microbiome were found to stimulate significant plant growth, and also prime plant defense against certain stresses [38]. Therefore, the current study was conducted to screen out the isolated indigenous heat tolerant bacteria with multiple plant growth

promoting activities, and evaluate the role of heat tolerant bacterium that reduced the negative effects of heat stress on growth and biochemical traits of tomato grown under greenhouse conditions.

## 2. Materials and Methods

### 2.1. Source and Growth Conditions of Bacteria

Soil samples were collected from 90 days-old tomato (cv. Riogrande) rhizosphere growing on a sandy loam in Larkana, Sindh, Pakistan (27.5570° N, 68.2028° E). The maximum average summer temperature of Larkana was 44 °C, while the minimum temperature was 29 °C. Soil properties of the sampling area were pH 8.8, electrical conductivity (EC) 0.38 dS m$^{-1}$ and organic matter content 0.83%. Bacterial strains were isolated using serial dilution method and plated on Luria-Bertani agar (LB) medium and incubated at 37 °C [39]. The pure cultures were obtained by picking the distinctive colonies after streaking at different dilution.

### 2.2. Heat Stress Tolerance Assay

The heat tolerance of *Bacillus cereus* was determined on population density basis at different ranges of temperature (ranging from 40, 45, 50, 55, and 60 °C) in LB medium. The bacterial strain was inoculated in 200 mL flasks (sterilized) that contained 100 mL LB medium, and kept at different temperature ranges (40, 45, 50, 55, and 60 °C) in shaking incubator with 120 rpm to analyze the bacteria. After 24 h of incubation, the optical density of inoculated bacterial culture was measured at λ = 600 nm using a spectrophotometer (Agilent 8453 UV–visible Spectroscopy System).

### 2.3. Biochemical Characterization of Bacteria

In Vitro Screening of Bacillus cereus for Plant Growth Promoting Traits

Phosphate solubilization test was conducted on PVK (Pikovskaya) agar plates according to the designed protocol of [40]. The bacterium was inoculated on Pikovskaya agar plates and incubated for seven days at 30 ± 0.1 °C.

The estimation of Indole-3-acetic acid production was conducted by colorimetric assay [41]. The production of ammonia was analyzed in peptone water according to the protocol of [42].

The bacterial isolate was screened for siderophore production using CAS (Chrome azurol S) agar media [43]. Hydrogen cyanide (HCN) production was conducted according to the method given by [44], streaking isolates on nutrient agar medium petri plates contained glycine (4.4 g/L). Whatman No. 1 filter paper was dipped in mix solution (0.5% picric acid solution and 2% sodium carbonate) and placed inside the plate lid, and incubation was completed at 30 ± 0.1 °C for four days. Production of HCN was indicated by the appearance of dark or light brown color.

### 2.4. Screening of Bacillus cereus for ACC Deaminase Activity

Qualitative and Quantitative Assays

Bacterial strain was checked for ACC-deaminase production as a nitrogen source [45]. Bacterial isolate was grown at 28°C for 24 h with continuous shaking at 120 rpm in 5 mL of Tryptic soy Broth (TSB) medium. The cells were centrifuged for 5 min at 3000 g, washed twice with distilled water, re-suspended in 0.1M Tris-HCl (pH 7.5). Spot inoculation of culture was done on petri plates which contain Dworkin and Foster salt media with and without ACC [46]. Plates with ammonium sulphate were considered as a positive control. Plates were placed on incubation for three days at 28 ± 2 °C. Growth on ACC-supplemented plates was compared with positive and negative control. Bacterial strain was further proceed for quantitative assay after the confirmation from qualitative activity. ACC quantification was done via measuring the alpha-ketobutyric acid produced by the cleavage of ACC by ACC-deaminase enzyme [47].

### 2.5. Exopolysaccharide Production (EPS)

The bacterium potential for EPS production was assessed following the method of Mu'minah et al. [48].

### 2.6. Extracellular Enzyme Assays

Sequential extracellular enzyme assays were implemented to comprehensively explore the bacterium potential. Protease test was conducted following the method of [49]. The amylase test was done by the method described by Ade [50]. The method of [51] was used for conducting pectinase test. Bacterial colony was collected, placed on microscopic slide and made smear with autoclaved loop. Few drops of 3% $H_2O_2$ was added and bubbles of gas formation was observed [52]. Formation of more no. of bubbles indicated the positive result for catalase and no bubble formation or small no. of dispersed bubbles showed the negative indication for catalase test.

### 2.7. Biochemical Characterization

Biochemical characteristics of *Bacillus cereus* were tested via microbial identification kits QTS-24 miniaturized recognition system (DESTO Laboratories, Karachi, Pakistan) as described by [53]. Bacterial culture was added into the QTS wells and incubation procedure was done for 24 h at 37 °C. After the incubation, reagents were added to the QTS wells containing bacterial strain as instructed in the manual of QTS-24 kit, and results were observed.

### 2.8. Screening for Antibiotic Resistance

The saturated disc diffusion method was used for detection of antibiotic resistance activity [54]. Strain (KTES) was added in LB broth and incubated at 37 °C. After two days, 100 µL of bacterial culture was spread onto plates containing LB agar. Discs, saturated with antibiotics, were spotted on to each plate. After incubation for 24 h, appearance of inhibition zones around the antibiotic discs were noted based on diameter of halo zone surrounding the disc. The strain was classified as resistant (<10 mm), intermediate (10 to 15 mm) or susceptible (>15 mm) to each antibiotic.

### 2.9. Quantification of Plant Growth Regulators

Extraction and quantification of IAA, gibberellins and kinetin from the bacterial isolate was performed following the method of [55]. Bacterial isolate was grown in King's B broth medium at normal and high temperature in shaking incubator (SHKE480HP, Thermo Fisher Scientific, Kansas City, United States of America; USA) at 150 rpm for 5 days. Bacterial culture (100 mL) was centrifuged at 15,000× *g* (Sorvall PC5C Plus centrifuge, Kendro Laboratory Products, USA) for 30 min. The supernatant was collected and pellet was discarded. The supernatant volume was reduced to 20 mL by evaporation with rotary evaporator (Yamato RE500, Yamato Scientific, Japan). The pH of the supernatant was adjusted to 2.8 with N 1HCl. The first step of extraction was done in centrifuge (IEC HN-SII, Thermo Fisher Scientific, USA) at 800× *g* for 5 min with ethyl acetate (1:1; *v/v*) for 3 times. The upper ethyl acetate phase was collected after every centrifugation. For the second step, the pH of water phase was adjusted to 7 with 1N NaoH and extracted with 0.4 v of water-saturated N-butanol for 3 times at 800× *g* in centrifuge for 5 min. The upper butanol phase was collected and water phase was discarded. Collected ethyl acetate and N-butanol phases were mixed and completely evaporated at 55 °C and diluted in 2.0 mL of MeOH:H3P04 (99.9:1; *v/v*). The diluted extract centrifuged at 12,000× *g* (Sorvall Biofuge Pico, Kendro Laboratory Products, Germany) for 10 min and complete removal of bacterial particles was done. The quantitative analysis of IAA, gibberellic acid (GA3) and kinetin production from bacterial culture was done with high performance liquid chromatography (HPLC) equipped with Waters 2487 Dual λ absorbance detector and column (Vydac 218Tp C18 5 µm). Peaks were detected at 214 nm. The analytical grade hormones (Sigma Aldrich, USA) were used as standards to identify IIA, GA3 and kinetin on chromatograms, and calculate their concentrations.

### 2.10. Phylogenetic Characterization of Bacterial Isolate

Extracted DNA was amplified with 16S rRNA genes [27F (5-AGAGTTTGATC AC TGGCTCAG-3) and 1492R (5-CGG CTTACCTTGTTACGACTT-3)]. The PCR products were sent to Macrogen, Korea, for commercial sequencing with universal 785F 16S rRNA gene specific primers. The sequences were assembled using Bio Edit software. The homology was determined by BLAST analysis of the consensus sequence in the NCBI (National Center for Biotechnology Information) database. Similar sequences were downloaded from NCBI database for phylogenetic tree construction. All the sequences obtained from NCBI database were aligned by muscle option in MEGA 6.0 software. Construction of phylogenetic tree was done in MEGA 6.0 software with 1000 bootstrap replicates.

### 2.11. Greenhouse Experiment

#### 2.11.1. Seed Sterilization and Inoculation with PGPR

Seeds of two tomato varieties (Riogrande and Sweetie) provided by Richter's Herbs, Goodwood, ON, Canada, were surface sterilized by soaking for 30 s in 70% ethanol. Seeds were further sterilized with 0.1% $HgCl_2$ for few seconds and then washed 3 times with sterile water and dried at room temperature [56]. The putative PGPR strain was grown in Luria Bertani broth media (LB media and composition of this media was 10 g Nacl (Sigma Aldrich), 10 g tryptone (Sigma Aldrich) and 5 g yeast extract (Sigma Aldrich) per liter) for 24 h and centrifugation was carried out at 3000 rpm for 10 min.

#### 2.11.2. Experimental Design and Setup

The investigation was carried out under greenhouse conditions at McGill University (Sainte-Anne-de-Bellevue, Canada) (45°24'27" N, 73°56'18"W) in a factorial completely randomized design having three replicates. The experiment included 4 treatments as follows: C = control (without bacterial inoculation/without heat stress), T1 = inoculated plant with *Bacillus cereus*, T2 = un-inoculated plant with heat stress and T3 = bacterial inoculated plants with heat stress. Tomato seedlings were sown in plug trays (53.5 × 25.5 cm) filled with autoclaved media (G-10: sand: manure, 2:1:1) having 5 seeds per cell. Seedlings were transplanted into 6-inch pots filled with autoclaved media and after 2 weeks of sowing, seedlings were transferred to greenhouse under semi-controlled conditions (70–C80% humidity, 25 ± 2 °C temperature and 14 h photoperiod: PAR 300 µmol $m^{-2}$ $s^{-1}$). Irrigation was supplied manually on daily basis and each pot was watered with 20 mL Hoagland solution (1.6 g/L) twice a week. Heat stress (42 °C) was applied at flowering stage to plants grown with and without bacterial inoculation. Plants were exposed to heat stress for 6 h/day in growth chamber till the fruiting stage. After exposure to heat stress, plants were placed again in greenhouse for recovery (temp. 25 ± 2 °C). Plants were harvested after 96 days of seed sowing. Harvested plants were washed thoroughly with sterile distilled water to remove the debris from roots. Harvested plants were preserved for further analysis.

#### 2.11.3. Plant Growth Traits

After harvesting, important plant growth traits were recorded. Root and shoot length was measured with a ruler, fresh weight was weighed with the help of an electronic balance, and leaf surface area was determined using a leaf area meter. Similarly, dry biomass was noted upon complete drying of plants in an oven for two days at 72 °C.

#### 2.11.4. Relative Water Content (RWC)

Weight of the leaf samples before and after oven drying were used for calculation of fresh biomass, dry biomass, and moisture contents, respectively. Leaf relative water content estimation was done with the method given by [57].

### 2.11.5. Effects of *Bacillus cereus* on Photosynthetic Pigments under Normal and Heat Stress Conditions

Chlorophyll *a, b* and carotenoids contents were extracted from tomato leaves (1 g) as described by [52]. Acetone (80%) was used to grind the leaf samples, and the grinded mixture was centrifuged at 10,000 rpm for 30 min. Chlorophyll *a, b* and carotenoids were observed in the supernatant at 645, 663 and 470 nm, respectively.

For protein content determination, protein extraction buffer (50 mM tris HCl buffer, pH 7.0, containing 3 mM MgCl2, 1 mM EDTA and 1.0% PVP, *w/v*) was used for lyophilization and halogenation of 1 g leaf samples. Centrifugation of samples was done for 20 min at 10,000 rpm at 4 °C. Protein (quantification) for each sample was described with the method of [58] and for standard, bovine serum albumin was used. The proline content determination was done by following the protocol of [58].

### 2.11.6. Antioxidant Enzyme Assays

Superoxide dismutase (SOD), Peroxidase (POD) and Catalase (CAT) activities were performed following the method of [59].

### *2.12. Statistical Analysis*

A two-way ANOVA was performed using Statistixs software (Version 8.1) for both varieties based on bacterial (control or inoculated) and temperature treatments (heat stress and non-heat stress). Adjustment for multiple comparisons were made using the LSD test, keeping significant level at $p \leq 0.05$. The application of bi-plots correlation analysis was performed on mean values of all variables using XL-STAT 2014.5.03.

## 3. Results

### *3.1. Bacterial Isolation and Screening for Heat Tolerance*

A total of 21 rhizosphere bacteria were isolated and purified. Isolates were screened for heat tolerance up to 60 °C. Out of 21 rhizosphere strains, 12 strains survived at 50 °C, 7 survived at 55 °C and only 2 survived at 60 °C. Only one strain was selected for experimentation due to its all positive plant growth promoting activities besides high temperature tolerance out of these two strains surviving at 60 °C (Table 1).

**Table 1.** Morphological and biochemical characterization of selected strain *Bacillus cereus* (KTES).

| Characteristics | Properties | | |
|---|---|---|---|
| **No. of bacterial isolates screened against heat stress** 21 | **50°C** 12 | **55°C** 7 | **60°C** 2 |
| **Morphological Attributes** | Rod shaped, colour: off white, form: circular, elevation: convex, margin: lobate, opacity: transparent, gram stain: + (positive), temperature range: 30–C60 °C | | |
| **Biochemical attributes** | **Positive for**: IAA, Phosphorous solubilization, ACC-deaminase, EPS production, Ammonia, catalase, amylase, pectinase and protease, CIT (sodium citratrate), Urea (urea hydrolysis), TDA (Tryptophan deaminase), ODC (Ornithine decarboxylase), H2S, IND (indole), GLU (Acid from glucose), MAL (Acid from maltose), SuC (Acid from sucrose), SORB (Acid from sorbitol), INOS (Acid from inositol), MEL (Acid from Melibiose), ADO (Acid from adonitol) and RAF (Acid from Raffinose). **Negative for**: Hydrogen cyanide, Siderophores, LDC (Lysine decarboxylase), MALO (sodium malonate), ONPG (ortho nitro phenyl β-D-galactopyranoside), ADH (Arginine dihydrolase), VP (Voges proskauer), GEL (Gelatin hydrolysis), MANN (Acid from mannitol), RHAM (Acid from Rhamnose) and ARAB: acid from arabinose. | | |

Morphological observation through naked eye and compound microscope; biochemical characterization of bacterial strain through QTS-24 kits and plant growth promoting activities.

### 3.2. Morphological-Physiological and Biochemical Traits of Selected Bacterial Strain

The selected bacterial strain was identified as gram positive and rod shaped via performing Gram staining procedure. Further, the bacterial strain was checked through microbial identification QTS-24 kits (DESTO Laboratories, Karachi, Pakistan). Morphological observations were done with the aid of compound microscope and naked eye, and biochemical characterization including plant growth promoting attributes were studied for the selected strain. Bacterial strain was found positive for catalase, amylase, pectinase and protease synthesis, whereas negative for hydrogen cyanide and siderophore production (Table 1).

Isolated strain exhibited ACC-deaminase activity and exopolysaccharide production under the normal and high temperature conditions. Bacteria showed the better growth on DF plates with and without nitrogen source. The bacteria on DF medium without ACC precursor showed less growth as compared to positive control and plates with ACC. Concentration of ACC and EPS was significantly higher in bacterial culture under the stressed condition as compare to normal condition (Figure 1).

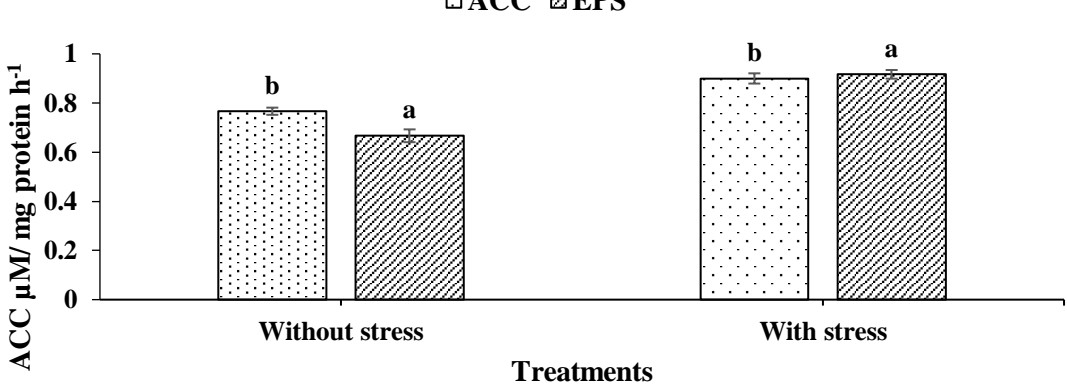

**Figure 1.** ACC-deaminase activity and exopolysaccharide production of *Bacillus cereus* KTES without stress (Control conditions and no heat) and stress condition (with heat). Letters indicate significant difference at $p < 0.05$ and error of the mean (n = 3) represented by bars.

The concentration of IAA was 0.55 and 0.44 μg/mL under normal and high temperatures, respectively. Gibberellic acid (GA) concentration under normal and high temperatures was 19.8 and 14.2 μg/mL, while kinetin concentration under normal and high temperatures was 43.6 and 25.1 μg/mL, respectively (Table 2). The bacteria expressed a wide range susceptibility level to antibiotics. These observations were made on the basis of diameter of zones around the antibiotics (Table 3).

**Table 2.** Quantitative assessments of hormone production of the selected putative PGPR *Bacillus cereus*.

| Indole Acetic Acid (μg/mL) | | Gibberellic Acid (μg/mL) | | Kinetin (μg/mL) | |
|---|---|---|---|---|---|
| NT | HT | NT | HT | NT | HT |
| 0.55 ± 0.026 | 0.44 ± 0.028 | 19.8 ± 1.18 | 14.22 ± 1.01 | 43.6 ± 17.3 | 25.18 ± 4.74 |

Hormone quantification in bacterial culture under normal and high temperature through High performance liquid chromatography, NT: Normal Temperature, HT: High Temperature.

**Table 3.** Antibiotic resistance activity of putative PGPR *Bacillus cereus*.

| Antibiotics | Diameter (mm) | Resistance Level |
|---|---|---|
| Erythromycin(E15) | 12 | Intermediate |
| Rifampicin (RD5) | 9 | Resistant |
| Ampicilin (AMP) | 3 | Resistant |
| Streptomycin (S10) | 8 | Resistant |
| Chloramphenolicum (C30) | 16 | Susceptible |
| Gentamycin (CN10) | 5 | Resistant |
| Fosomycin (FOS 50) | 12 | Intermediate |
| Spectinomycin (SH25) | 6 | Resistant |
| Neomycin (N10) | 15 | Intermediate |
| Tetracyclin (TE 30) | 2 | Resistant |
| Lincomycin (My15) | 9 | Resistant |
| Clindamycin (DA2) | 8 | Resistant |
| Penicillin (P10) | 10 | Intermediate |
| Kanamycin (K30) | 16 | Susceptible |

Resistant: <10 mm, intermediate: 10–C15 mm, Susceptible: >15.

### 3.3. Strain Identification and Accession Number

The studied strain was identified as *Bacillus cereus* after blast on NCBI and phylogenetic tree constructed using MEGA 6.0 software (Figure 2). Maximum Composite Likelihood method was used to compute the evolutionary distances. Analysis for evolutionary history was carried out with MEGA 6.0. Studied strain was submitted to NCBI with accession number of MK784894.

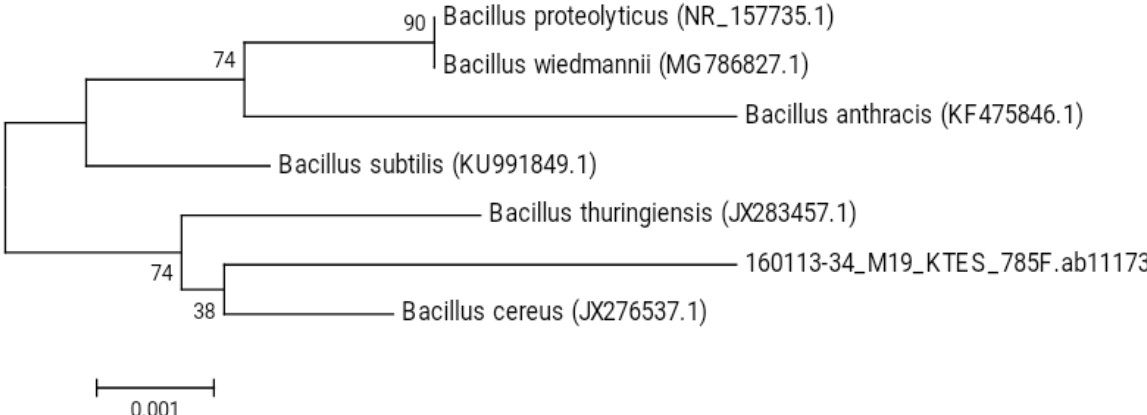

**Figure 2.** The evolutionary history was constructed by Neighbor-Joining method. Sum of branch length of tree was 0.01959378. Bootstrap test—800 replicates.

### 3.4. Response of Plant Biomass to the Inoculation of Bacillus cereus

Bacterial inoculation under normal condition (T1) increased the shoot length by 23.7% and 20.9% for Riogrande and Sweetie compared to the un-inoculated control (C) under normal conditions, whereas heat stress reduced shoot length by 65.8% and 65.3% in Riogrande and Sweetie in comparison to un-treated control, respectively (Table 4). However, bacterial inoculation under heat stress condition (T3) enhanced the shoot length by 53.1% and 45.05% for Riogrande and Sweetie, respectively, compare to un-inoculated heat stress treatment (T2). Inoculation of bacteria increased root length under normal condition (T1) in both varieties by 31.9% (Riogrande) and 27.4% (Sweetie) compare to un-treated control treatment (Table 4). Likewise, the T3 (bacterial inoculation under heat stress) relieved the root length by 50.3% (Riogrande) and 48.5% (Sweetie) in comparison to T2 (without bacteria under heat stress). *Bacillus cereus* (T1) up-regulated the accumulation of fresh and dry biomass in Riogrande and Sweetie by 27.9%, 38.9% and 27.1%; 6.4%, accordingly when compared with T2. Treatment T3

helped the plants in achieving more fresh and dry biomass by 62% and 45.2% (Riogrande), and 41.2%, 42.9% (Sweetie), respectively, relative to T2 (Table 4). Similar trend with regard to leaf surface area was observed where it was enhanced in Riogrande (30.2%) and Sweetie (19.9%) under T1 treatment in contrast to un-treated control. *Bacillus cereus* extended the leaf surface area (53.4% for Riogrande and 46.3% for Sweetie) under heat stress in T3 as compared to T2 (Table 4).

The number of flowers increased with bacterial inoculation in both the varieties (Riogrande and Sweetie) under both normal and heat stress conditions in contrast to un-inoculated treatments (Table 5). For Riogrande, the number of flowers was reduced by 49.7% under heat stress without inoculation of bacteria (T2) compared to control (C). Under normal temperatures, inoculation of bacteria (T1) increased the flower number by 46.5% compare to the control (C) for Riogrande but under high temperature conditions, the number of flowers increased by 54.1% in comparison to un-inoculated plants that was grown under heat stress (T2) (Table 5). On the other hand for Sweetie, flower number for un-inoculated plants under heat stress (T2) was decreased by 62.4% compared to the control (C) and an increase of 53.5% was observed upon inoculation of bacteria under normal condition (T1) in comparison to control (C). Bacterial inoculation under heat stress (T3) increased the number of flowers by 14.7% compared to control and 67.9% compared to un-inoculated plants under heat stress (T2) (Table 5).

It was observed that bacterial treatment under normal and heat stress conditions (T1 and T3) produced a greater number of fruits compared to control (C) and un-inoculated plants under heat stress (T2). The un-inoculated plants of Riogrande grown under heat stress (T2) showed a 56.5% reduction in fruit number compare to control (C), and inoculated plants under normal condition (T1) showed an increase of 48.9% in fruit number compared to the control (C). Bacterial inoculation under heat stress (T3) produced 46.3% and 76.7% higher fruit numbers in Riogrande compared to the control (C) and un-inoculated plants under heat stress (T2), respectively (Table 5). The un-inoculated Sweetie plants under heat stress (T1) produced 56% lesser fruits compared to control (C), and the inoculated plants under normal condition (T2) showed 57.5% more fruits compared to the control (C). Bacterial inoculation under heat stress increased fruit number by 13.75 and 62.1% as compared to control (C) and un-inoculated sweetie plants under heat stress condition (T1), respectively (Table 5).

**Table 4.** Effect of bacterial isolates *Bacillus cereus* on growth parameter of two tomato varieties (Riogrande and Sweetie) under heat stress and bacterial inoculated treatment.

| Treatments | SL (cm) | | RL (cm) | | FW (g) | | DW (g) | | LSA (m2) | |
|---|---|---|---|---|---|---|---|---|---|---|
| | Riogrande | Sweetie | Riogrande | Sweetie | Riogrande | Sweetie | Riogrande | Sweetie | Riogrande | Sweetie |
| Control | 50 ± 1.15c | 45.5 ± 0.43d | 15.4 ± 0.46bc | 12.4 ± 0.49d | 39.7 ± 1.07c | 34.3 ± 0.58d | 11.0 ± 1.11c | 11.3 ± 0.38c | 27.2 ±1.58c | 25.5 ± 0.84cd |
| T1 | 65 ± 2.08a | 57.3 ± 0.41b | 22.6 ± 1.21a | 17.1 ± 0.41b | 55.1 ± 1.21a | 47.1 ± 1.02b | 18.0 ± 0.42b | 21.1 ±1.85a | 39.0 ± 0.96a | 31.9 ± 0.94b |
| T2 | 17.1± 0.99g | 15.7 ± 0.437g | 7.6 ± 0.40e | 7.66 ± 0.28e | 11.2 ± 0.34g | 12.8 ± 0.60g | 4.1 ± 0.46e | 8.96 ± 0.26cd | 10.7 ± 0.34f | 9.16 ± 0.31f |
| T3 | 36.5 ± 1.53e | 28.63 ± 0.81f | 15.3 ± 1.09bc | 14.9 ± 0.43c | 29.5 ± 0.52e | 21.8 ± 0.32f | 7.1 ± 0.43d | 15.7 ± 0.89b | 22.6 ± 1.90d | 17.1 ± 0.95e |

Growth traits were measured at 96 days after seed germination under SL (shoot length), RL (root length), FW (fresh weight), DW (dry weight) and LSA (leaf surface area). Values (for the two varieties and different treatment) without a common letter are significantly differed ($p > 0.05$). T1- with bacteria, T2- with heat, T3- heat stress coupled with bacterial inoculation.

**Table 5.** Effects of *Bacillus cereus* on the flowers and fruits of tomato varieties under heat stress.

| Treatments | Flower Number | | Fruit Numbers | |
|---|---|---|---|---|
| | Riogrande | Sweetie | Riogrande | Sweetie |
| Control | 14.6 ± 1.45b | 13.3 ± 1.45b | 7.66 ± 0.88cd | 8.33 ± 0.88cd |
| T1 | 27.3 ± 2.72a | 28.6 ± 3.28a | 15 ± 4.50 ab | 19.6 ± 0.881a |
| T2 | 7.33 ± 0.88c | 5 ± 1.73c | 3.33 ± 0.33d | 3.66 ± 1.45d |
| T3 | 7.33 ± 0.88b | 15.6 ± 2.02b | 14.3 ± 1.45ab | 9.66 ± 0.33bc |

Flowers and fruits no. of both varieties observed under normal and heat condition. T1- with bacteria, T2- with heat, T3- with bacteria, and heat values (for the two varieties and different treatment) without a common letter are significantly differed ($p > 0.05$).

*3.5. Relative Water Content*

Lower leaf water potential was observed in un-inoculated plants under normal and heat stress conditions in both varieties (Riogrande and Sweetie). Bacterial inoculation (T1) enhanced water content by 28.6% and 37.3% in Riogrande and Sweetie, respectively, as compared to control, while T3 raised the water content by 33.4% for Riogrande and 41.4% for Sweetie compared to T2 (Table 6).

*3.6. Chlorophyll Contents*

Chlorophyll *a, b* and carotenoid contents decreased in response to heat stress (T2) compared to control (C) wherein bacterial inoculation, significantly stimulated chlorophyll a, b and carotenoid biosynthesis in both tomato varieties under heat stress and normal conditions (Table 6). Chlorophyll a, b and carotenoid contents of Riogrande augmented by 33%, 20.3% and 61.4%, respectively, with inoculation under normal condition (T1) compared to control (C) (Table 6). Inoculation of bacteria under heat stress (T3) also stemmed up chlorophyll *a, b* and carotenoid concentrations of Riogrande by 54.7%, 59.5% and 64.4%, respectively, in comparison to un-inoculated plants with heat stress (T2) (Table 6). Similarly, in the variety Sweetie, inoculation with bacteria (T1) increased chlorophyll *a* (51.4%), *b* (14.2%) and carotenoid (56.1%) contents under normal temperature conditions (T1) compared to control (C) while inoculation of bacteria under heat stress (T3) enhanced the chlorophyll *a, b* and carotenoid by 74.6%, 66.9% and 79.4%, respectively, in contrast to T2, i.e., un-inoculated plants under heat stress condition (Table 6).

*3.7. Protein and Proline Contents*

Application of bacteria to both varieties (Riogrande and Sweetie) enhanced protein and proline contents with heat stress conditions and vice versa (Table 7). For Riogrande, inoculation with bacteria under normal conditions (T1) induced the protein synthesis by 36.8% and proline synthesis by 25.1% compared to control (C). Application of bacteria under heat stress (T3) also heightened the leaf protein content by 59.4% and proline content by 32.6% with respect to un-inoculated Riogrande plants under heat stress (T2) (Table 7). For Sweetie, bacterial inoculation under normal temperature (T1) increased protein concentration by 54% and proline concentration by 34.4% compared to control (C). Protein and proline contents were enhanced by 52.9% and 6%, accordingly, in inoculated Sweetie plants under treatment T3 upon comparison with plants under T2 (Table 7).

*3.8. Antioxidant Activities*

Antioxidant enzymes such as superoxide dismutase (SOD), peroxidase (POD) and catalase (CAT) were significantly up-regulated in response to heat stress as well as application of bacterial inoculum (Table.7). The results showed that antioxidant activities of Riogrande tomato plants exhibited 62%, 61.5%, and 82.3% higher SOD, POD and CAT activities, respectively, for un-inoculated plants grown under heat stress (T2) compared to control (C) (Table 7). Likewise, bacterial inoculation under normal conditions (T1) raised POD activity by 97.3% and CAT activity by 31.8% compared to control (C). Bacterial inoculation under heat stress conditions (T3) also increased SOD, POD and CAT activities by 42.9%, 96.2% and 11.3%, respectively, with respect to un-inoculated Riogrande plants under heat stress (T2) (Table 7). For Sweetie plants, the antioxidant activities strengthened by 63.3%, 58.3% and 86.7% for SOD, POD and CAT, accordingly, when un-inoculated tomato plants were grown under heat stress conditions (T2) compared to control (C). However, bacterial inoculation under normal conditions (T1) significantly strengthened POD activity by 96.2% and CAT activity by 35.2% compared to control (C). Application of bacteria under heat stress conditions (T3) to tomato plants produced significantly more SOD, POD and CAT amounting to 40.3%, 95.7% and 29.5% higher, respectively, as compared to un-inoculated tomato plants under heat stress conditions (T2) (Table 7).

### 3.9. Correlation Analysis

The bi-plots correlation analysis disclosed that *Bacillus cereus* had a positive effect on growth of both tomato varieties under heat stress conditions. Correlation between traits is presented with red and blue dots which refer to the correlation between treatments. The variables were present in the same quadrant which were very close to each other, and were strongly and positively correlated. The combine correlation bi-plot between F1 and F2 revealed 92.45% variation in which F1 contributed 72.39% and 20.06% for F2. There is existence of a significant positive correlation (alpha = 0.05) between shoot and root length; chlorophyll a, b and carotenoids; protein contents; no. of flowers, no. of fruits; fresh weight and dry weight. However, there is a negative correlation in the case of SOD, proline and catalase (Figure 3).

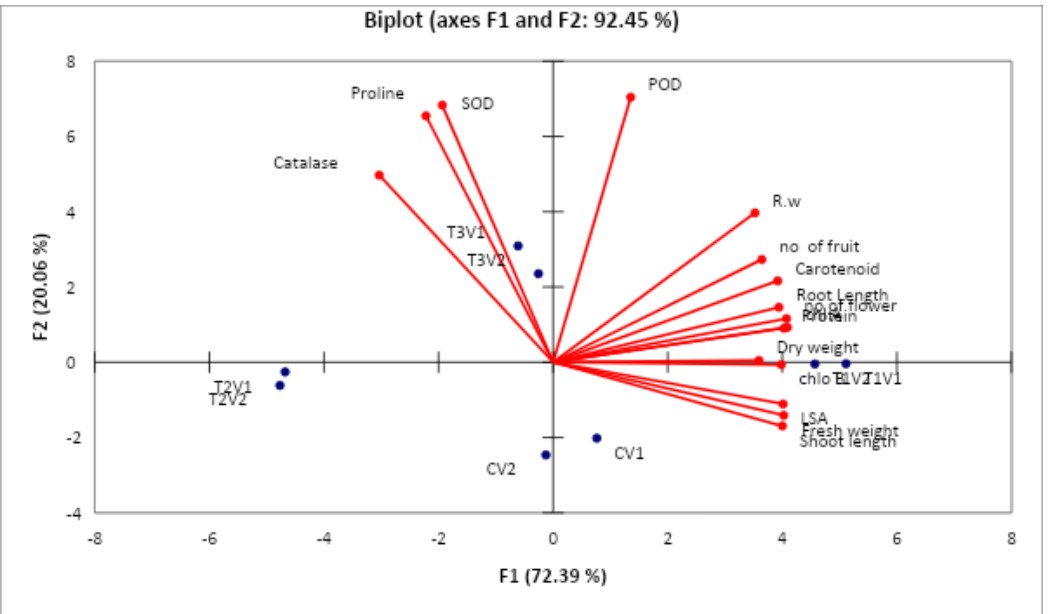

**Figure 3.** The correlation biplot analysis between different treatments, tomato varieties, growth traits and biochemical analysis.

**Table 6.** Influences of *Bacillus cereus* on chlorophyll contents in leaves under heat stress.

| Treatments | Chlorophyll a | | Chlorophyll b | | Carotenoid | | Relative Water Content | |
|---|---|---|---|---|---|---|---|---|
| | Riogrande | Sweetie | Riogrande | Sweetie | Riogrande | Sweetie | Riogrande | Sweetie |
| Control | 1.76 ± 0.03b | 1.38 ± 0.09c | 1.53 ± 0.01c | 1.44 ± 0.02d | 7.83 ± 0.10e | 9.25 ± 0.03d | 53 ± 1.73d | 43.6 ± 1.76e |
| T1 | 2.63 ± 0.06a | 2.84 ± 0.06a | 1.92 ± 0.02a | 1.68 ± 0.024b | 20.3 ± 0.63a | 21.1 ± 0.50a | 74.3 ± 1.20a | 69.6 ± 0.88ab |
| T2 | 0.66 ± 0.02d | 0.48 ± 0.14d | 0.55 ± 0.01f | 0.41 ± 0.05g | 4.19 ± 0.02f | 2.73 ± 0.02g | 44.3 ± 2.60e | 35.3 ± 1.45f |
| T3 | 1.46 ± 0.04c | 1.89 ± 0.11b | 1.36 ± 0.03d | 1.24 ± 0.023e | 11.8 ± 0.28c | 13.3 ± 0.26b | 66.6 ± 0.88bc | 60.3 ± 0.33cd |

T1- with bacteria, T2- with heat, T3- with bacteria and heat. Values (for the two varieties and different treatment) without a common letter are significantly differed ($p > 0.05$).

**Table 7.** Influence of *Bacillus cereus* on antioxidants (POD, SOD, and CAT) production, proline and protein accumulation in the leaves of tomato varieties of Riogrande and sweetie under normal and heat stress.

| Treatments | POD ($\mu$molg$^{-1}$ FW min$^{-1}$) | | SOD ($\mu$molg$^{-1}$ FW min$^{-1}$) | | CAT ($\mu$molg$^{-1}$ FW min$^{-1}$) | | Proline ($\mu$molg$^{-1}$ FW min$^{-1}$) | | Protein ($\mu$molg$^{-1}$ FW min$^{-1}$) | |
|---|---|---|---|---|---|---|---|---|---|---|
| | Riogrande | Sweetie | Riogrande | Sweetie | Riogrande | Sweetie | Riogrande | Sweetie | Riogrande | Sweetie |
| Control | 0.03 ± 0.008g | 0.05 ± 0.008Fg | 2.13 ± 0.03e | 1.64 ± 0.02g | 0.03 ± 0.008e | 0.02 ± 0.002f | 24.4 ± 1.25f | 21.3 ± 1.09g | 3.53 ± 0.03c | 2.23 ± 0.03e |
| T1 | 1.12 ± 0.01d | 1.35 ± 0.02c | 1.87 ± 0.008f | 1.53 ± 0.027g | 0.44 ± 0.01d | 0.34±0.002e | 32.6 ± 0.42d | 28.1 ±0.43e | 5.50 ± 0.05a | 4.85 ± 0.04b |
| T2 | 0.08 ± 0.008ef | 0.12 ± 0.01e | 5.61 ± 0.01c | 4.47 ± 0.026d | 1.71 ± 0.01b | 1.66 ± 0.03b | 46.1 ±0.65c | 48.5 ± 0.82c | 1.14 ± 0.03g | 1.61 ± 0.08f |
| T3 | 2.13 ± 0.0b | 2.82 ± 0.02a | 8.02 ± 0.03a | 7.49 ± 0.19b | 1.93 ± 0.02a | 1.17 ± 0.02c | 68.4 ± 0.55a | 51.6 ± 0.76b | 2.81 ± 0.01c | 3.40 ± 0.04d |

The effect of inoculated bacteria and heat stress, POD (peroxidase), SOD (superoxide dismutase), CAT (catalase), Proline and protein. T1- with bacteria, T2- with heat, T3- with bacteria and heat. Values (for the two varieties and different treatment) without a common letter are significantly differed ($p > 0.05$).

## 4. Discussion

Plant growth promoting rhizobacteria demonstrated a prominent role in plant health improvement and a deliberate greater level of tolerance to various important biotic and abiotic stresses. Global warming is a result of greenhouse gases emission into the atmosphere that cause the serious issues of sustainability [60–62]. The isolated best performing bacterial strain was screened to determine its heat tolerance capability and plant growth potential under normal and heat stress conditions. The current study describes the potential of heat tolerant PGPR *Bacillus cereus* for the production of ACC-deaminase, EPS, extracellular enzymes activities that alter the growth traits of tomato plant under heat stress. There is a wide range of variation in the bacterial resistance while tested against different antibiotics. Variation in bacterial resistance reaction against the tested antibiotics might be possible because of bacteria growth in different environmental conditions and the way of exposure of PGPR to stress conditions [63]. Up to the best of our knowledge, this is the very first attempt of reporting the evidence for minimizing the effects of high temperature stress in tomato with the application of *Bacillus cereus* during the current investigation.

Ethylene concentration increased in plant tissues due to various abiotic and biotic stresses [64,65]. Threshold production of ethylene during abiotic stress reduced seed germination and root development that stunt the plant's growth. ACC-deaminase producing *Bacillus cereus* helped to alleviate ethylene production which cleaved the ACC to α-ketobutyrate and ammonia, and decreased the adverse effects of ethylene on plant growth under heat stress. The inoculation of ACC- deaminase-producing bacteria in plants has been linked with abiotic stress tolerance [66]. It also enhanced the nutrient uptake and root growth [67]. *Bacillus cereus* showed the potential for exopolysaccharides (EPS) production under normal and stress conditions. Sandhya et al, 2009b [30], reported that EPS- producing bacteria has the ability to provide resistance to plants against abiotic stress. Quantification of plant growth regulators (IAA, GA, and kinetin) and plant growth promoting activities of *Bacillus cereus* during this research work were strengthened with the findings of [68]. Our findings were further supported by the results of [69,70]; different bacterial strains (*Aeromonas punctata*, *Serratia marcescens* and *Azospirillum brasilense*) has improved the growth and induced morphological alterations in *Arabidopsis thaliana*. Production of gibberellin by PGPR strain *Bacillus cereus* and its effects on tomato varieties was supported with the results of [71], as they stated that inoculation of gibberellins-producing strain *Promicromonospora* sp. (SE188) increased gibberellins concentration in plant shoots. Demonstration of [72,73] supported our findings that the inoculation of cytokinin- producing bacteria enhanced the shoot growth and fruit formation, and increased the resistance of plants to abiotic stress. Moreover, our findings were strengthened with the results of [74], that the resistance of *Platycladus orientalis* to abiotic stress increased with cytokinin-producing *Bacillus subtilis*.

PGPR inoculation observed to minimize the adverse effects of heat stress upon the plant growth and its productivity [75]. Tomato variety Sweetie was comparatively more heat-sensitive than Riogrande under inoculated conditions. Daim et al, 2014, [76] evaluated two wheat varieties under heat stress conditions and revealed that the application of PGPR improved its growth and supported our results. In the current study, bacterial application reduced the negative effects of heat stress on all measured growth traits of both tomato cultivars. Bacterial inoculation under heat stress increased the biomass of both varieties compared to un-inoculated plants. Heat stress mitigation was also observed in sorghum plants through bacterial application [77]. Chandra et al, 2018, [78] found that inoculation of finger millet with a *Pseudomonas* sp increased growth traits, fresh weight, dry weight and shoot length, and root length under abiotic stress and normal conditions. Previous studies demonstrated that growth of plants increased in response to PGPR application because of the production of plant growth regulators inside roots which stimulates root development and maximizes water and nutrient absorption from soil [75]. Flowering and fruit sets have been influenced by high temperatures in tropical and temperate regions. The number of flowers can be reduced with exposure to heat stress as reported previously [4]. Poor fruit setting has also been associated with low levels of carbohydrates and growth regulators which released sink tissues in plants due to elevated temperature [79]. The results of our current

study showed that the number of flowers and fruits was increased by the inoculation of bacteria in two tomato varieties under heat stress and normal conditions in comparison to varieties without bacterial application.

Our results showed that chlorophyll *b* content was higher in heat stress as compared to normal conditions in Riogrande. Increased chlorophyll contents could be due to higher photosynthetic leaf area that results from inoculation with PGPR, which was reduced for un-inoculated plants exposed to heat stress compared to un-inoculated plants grown under normal conditions [80,81]. Furthermore, our current results are strongly supported by the findings of [82], as they reported that application of P. putida enhanced the chlorophyll content in the shoots of canola plant. Similarly, [83–85] also documented that application of *Brevibacterium* sp (FAB3) helped to mitigate abiotic stress conditions via enhanced chlorophyll content and lead to the improvement of plant yield attributes.

Antioxidants served as an indicator to plant tolerance under abiotic stresses. Low molecular weight enzymes (SOD, POD and CAT) produced by plants confer the plant tolerance under stress conditions [86]. The concentration of antioxidant enzymes increased in both varieties under heat stress and normal conditions with bacterial inoculation compared to control. Antioxidant activities provided the potential to plants against stress by preventing the production of reactive oxygen species that caused the oxidative damage and also enhanced the proline concentration in plant tissues [83]. Reactive oxygen species are scavenged by SOD activity which was increased in response to bacterial treatments. The mechanism of antioxidant enzymes like SOD and POD convert O-2 to H2O2 and the resulting substrate is removed by CAT [58]. Accumulation of proline in leaves is an adaptive mechanism that regulated membrane permeability in cells and influenced water movement between tissues under heat stress conditions. Proline concentration was higher in both bacterial inoculated tomato varieties compared to un-inoculated tomato varieties under heat stress and normal conditions. Results of Grover et al, [1] demonstrated that the level of proline is significantly increased by inoculation of heat tolerant bacteria *P. putida*. Moreover, [87,88], described that *P. putida* significantly improved the accumulation of proline in wheat under heat stress. Our findings are further strengthened by [58], as they observed that plant metabolites (protein and proline) get accumulated in sorghum seedlings under heat stress upon inoculation with bacteria. Current investigation demonstrated the possible mechanisms of heat tolerant PGP, ACC-deaminase and EPS producing *Bacillus cereus* that induced the tolerance against heat stress in tomato varieties.

## 5. Conclusions

Inoculation of thermo-tolerant plant growth promoting rhizobacteria (PGPR, *Bacillus cereus*) could be an effective strategy for alleviation or minimizing the negative effects of heat stress on plant growth and biochemical traits of wheat. The role of heat tolerant PGPR, ACC-deaminase and EPS producing *B. cereus* strain revealed better improvement in the physiological (shoot and root length, fresh and dry weight, and leaf surface area) and biochemical traits (chlorophyll contents, relative water content, protein, proline, and antioxidant activities) of two tomato varieties grown under heat stress. The identified bacterial strain possessed the plant growth promoting activities that include IAA production and phosphate solubilization, which could adequate the damages due to stress and sustain the growth and plant health. Current findings revealed that the microorganisms such as *Bacillus cereus* play a possible role in mitigation of negative effects of heat stress on crop growth and development, and this approach may lead to production of microbial products for reduction of such effects. The use of beneficial soil microorganisms owed to a rise in demand of environment safety and food security. Current research work opens the opportunities to assess the possible role of bio-inoculants in minimizing the heat stress problem in plants in field condition.

**Author Contributions:** Conceptualization, T.M., D.S., S.A., M.F.S. and H.J.C.; data curation, S.u.R., A., A.A.A. and M.A.O.S.; formal analysis, T.M., T.S., M.F.S. and A.A.; funding acquisition, M.F.S., A.A.A. and A.A.I.; investigation, D.S., M.F.S., A.A.A., S.A. and H.J.C.; methodology, T.M. and T.S.; project administration, D.S, A.A.A., H.J.C. and M.A.O.S.; resources, M.F.S., A.A.A., A.A.I. and M.A.O.S.; software, S.u.R. and A.; supervision, S.A. and

H.J.C.; validation, T.H.I.S. and A.A.I.; visualization, T.S. and T.H.I.S.; writing—original draft, T.M. and S.u.R.; writing—review and editing, S.u.R., D.S., M.F.S., A.A.A. and S.A. All authors have read and agreed to the published version of the manuscript.

**Funding:** The authors extend their appreciation to the Deanship of Scientific Research at King Saud University for funding this work through research group no. (RGP-1438–011). The research was funded by ALP project (CS-374), Pakistan Agricultural Research Council Islamabad and McGill University.

**Acknowledgments:** The authors extend their appreciation to the Deanship of Scientific Research at King Saud University for funding this work through research group no. (RGP-1438–011). Authors are thankful to Rachel Backer, Plant Science Department, Macdonald campus, McGill University, for technical and English editing of manuscript.

**Conflicts of Interest:** The authors declare no conflict of interest.

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
