# Peer review of "Mitigation of Heat Stress in Solanum lycopersicum L. by ACC-deaminase and Exopolysaccharide Producing Bacillus cereus: Effects on Biochemical Profiling"

_sustainability, doi:10.3390/su12062159_

Round 1

Reviewer 1 Report

The authors with this new submission, improved the previously submitted manuscript, following the reviewers' suggestions.

Reviewer 2 Report

Dear Authors,

Please check following points.

L341 - Should be Table 6.

L378 - Should be Table 7.

Reviewer 3 Report

I believe that the manuscript is of enough quality to be published in the present form. 

This manuscript is a resubmission of an earlier submission. The following is a list of the peer review reports and author responses from that submission.

Round 1

Reviewer 1 Report

The present paper entitles "Mitigation of Heat Stress in Solanum lycopersicum L. by ACC-deaminase and Exopolysaccharide Producing Bacillus cereus: Effects on Biochemical Profiling" investigates the potential of Bacillus cereus in mitigating heat stress effects in tomato. Interestingly, the authors reported that bacterial inoculation had valuable effects on physiological and biochemical parameters under heat stress conditions and this would be important for the development of heat-tolerant strategy for crop production. The paper is well organized, the results are clearly reported and discussed.

Reviewer 2 Report

Reference: Mukhtar et al, “Mitigation of Heat Stress in Solanum lycopersicum L. 2 by ACC-deaminase and Exopolysaccharide Producing 3 Bacillus cereus: Effects on Biochemical Profiling

This is an interesting work dealing with a heat-resistant Rhizobacteria with promoting growth properties over tomato. In a nutshell, the authors recovered from rhizosphere tomato microbiome a heat-tolerant isolate and characterized biochemically; reaching the conclusion it was a Bacillus cereus strain. Then, they inoculated with this bacterium two tomato lines, and measured a series of physiological and biochemical traits to show the inoculation increase growth of tomato under both control and heat conditions. The main drawback of this work, in my opinion, is the use of a single bacterial strain and the lack of proper inoculation controls, as discussed below.

First, to properly demonstrate that a heat tolerant strain display heat-related effects over tomato, at least one heat-sensitive strain should have been used in parallel. Testing many different strains would be ideal, to give the reader the idea that a proper repertoire of alternative hypothesis are been tested (as good science always should; a sample of one is no valid evidence for any kind of bias). I certainly think the authors missed the opportunity to develop some sort of quick bio-screen to recognize which of the initial isolates display promoting effects, and which doesn’t.

Second, the controls for inoculation were non-inoculated plants. However, ideally, another two added controls ought to have been used: (1) a positive control inoculated with a different species of known promoting effects (which could also help decide if the new isolate is in any measure better than previous ones, and also if heat-related effects could be simply indirectly due to better growth with no direct link to heat, as I suspect in the case under study here); and (2) also a negative control inoculated with a strain displaying no promoting growth effects, to demonstrate that the inoculation per se does not alter plant responses (e.g. plant-microbe interaction responses, or the use of inoculating media sourcing extra nutrients to plants).

Reviewer 3 Report

Dear Authors,

There are many abbreviations in the abstract. As an example L28 - ACC-deaminase (1-aminocyclopropane-1-carboxylate). L32 -typo in morph-physiological parameters. L34 - EPS production -Extracellular polymeric substances. PGPR - plant growth-promoting rhizobacteria.This abstract is not following the scientific writing standards. Please elaborate abbreviations and follow scientific writing style. As an example, please refer 

Characterisation of plant growth-promoting rhizobacteria from rhizosphere soil of heat-stressed and unstressed wheat and their use as bio-inoculant by Asraf et al. 2019.

Bioprospecting Plant Growth-Promoting Rhizobacteria That Mitigate Drought Stress in Grasses. Jochem et al. 2019.

L53 - Heat stress is a problem to agriculture feild.....It is true but you should add suitable reference.

What is proline transport in line 61? Suddenly appeared it. Rhizobacteria has several genera and highly diverse. Your key component is this bacteria but simply explains it by one sentence. Line 68, again lytic enzymees - what is the role of lytic enzymes. There is no proper linkage between your ideas. We have to refer by ourselves to undestand the function of them in respect to your topic. Line 75 - ACC producing bacteria have the ability to supply nitrogen and energy to plants. Who said so? Where is the reference?

L94 550C typo

L109 DF plates?

Please add a photograph bacterial growth on DF plates as a supplimentary figure under several conditions as you described. 

Figure 1 - Which kind of mean comparison test that you have used? Did you compare ACC and EPS as well? Usually letter a should be in the error bar of highest colum. So letters should be exchanged. You have said about DF medium? how it comes then LB medium?

Line 119- GA , Gibberellic acid?How many replicates have you used for hormone measurements?

L140 - Riogrande and sweetie suddnely appeared.

Please rewrite the manuscript by following scientific writing.

Reviewer 4 Report

Mukhtar reported an interesting study regarding the use of PGPR to improve tomato tolerance to heat stress.

The manuscript is well written and easy to read; however, some points should be better addressed.

Lines 59-60. This sentence regards all crops or only tomato? In my experience until 25 °C there are not problems in tomato flowering. Please specify and add a reference.

Lines 74-75. The authors should add a reference.

Line 84. The authors should consider also this reference https://doi.org/10.1094/PBIOMES-06-19-0028-R

Line 80. The authors should consider also this reference https://doi.org/10.3390/horticulturae5040079

Line 336. The authors should report the name of the cultivar, the main agronomic techniques used in the production, e.g. fertilizers and plant protection products used as well as the herbicides used. Also the physical and chemical soil properties and the weather conditions are fundamental for this kind of work.

Line 342. Maybe 60 °C?

Line 344. Temperature ranges? Which ones?

Line 439. The authors should add the solar radiation or PAR used in the greenhouse

Line 443. The authors should add all the information regarding the growing media, providers, chemical composition

Lines 294-295. This sentence is in contrast with your affirmation about the originality of your work, “results who demonstrated that bacterial application alleviated heat stress in sorghum plants [60]”